# Endoscopic Ultrasound to Identify the Actual Cause of Idiopathic Acute Pancreatitis: A Systematic Review

**DOI:** 10.3390/diagnostics13203256

**Published:** 2023-10-19

**Authors:** Francesco Cammarata, Lucrezia Rovati, Paola Fontana, Pietro Gambitta, Antonio Armellino, Paolo Aseni

**Affiliations:** 1Department of General Surgery, Ospedale Luigi Sacco, Università degli Studi di Milano, 20157 Milan, Italy; fra.cammarata@hotmail.it; 2School of Medicine and Surgery, University of Milano-Bicocca, 20126 Milan, Italy; 3Emergency Department, ASST Grande Ospedale Metropolitano Niguarda, 20162 Milan, Italy; 4Department of Gastroenterology, ASST Ovest Milanese, 20025 Legnano, Italy; paola.fontana@asst-ovestmi.it (P.F.); pietro.gambitta@asst-ovestmi.it (P.G.); 5Endoscopy Division, Ospedale San Leopoldo Mandic di Merate, ASST Lecco, 23807 Lecco, Italy; a.armellino@asst-lecco.it; 6Department of Biomedical and Clinical Sciences “L. Sacco”, Università degli Studi di Milano, 20157 Milan, Italy

**Keywords:** acute pancreatitis, idiopathic acute pancreatitis, echoendoscopy, endosonography, endoscopic ultrasound, microlithiasis, biliary pancreatitis, bile duct stones

## Abstract

Idiopathic acute pancreatitis (IAP) presents a diagnostic challenge and refers to cases where the cause of acute pancreatitis remains uncertain despite a comprehensive diagnostic evaluation. Endoscopic ultrasound (EUS) has emerged as a valuable tool in the diagnostic workup of IAP. This review explores the pivotal role of EUS in detecting the actual cause of IAP and assessing its accuracy, timing, safety, and future technological improvement. In this review, we investigate the role of EUS in identifying the actual cause of IAP by examining the available literature. We aim to assess possible existing evidence regarding EUS accuracy, timing, and safety and explore potential trends of future technological improvements in EUS for diagnostic purposes. Following PRISMA guidelines, 60 pertinent studies were selected and analysed. EUS emerges as a crucial diagnostic tool, particularly when conventional imaging fails. It can offer intricate visualization of the pancreas, biliary system, and adjacent structures. Microlithiasis, biliary sludge, chronic pancreatitis, and small pancreatic tumors seem to be much more accurately identified with EUS in the setting of IAP. The optimal timing for EUS is post-resolution of the acute phase of the disease. With a low rate of complications, EUS poses minimal safety concerns. EUS-guided interventions, including fine-needle aspiration, collection drainage, and biopsies, aid in the cytological analysis. With high diagnostic accuracy, safety, and therapeutic potential, EUS is able to improve patient outcomes when managing IAP. Further refinement of EUS techniques and cost-effectiveness assessment of EUS-guided approaches need to be explored in multicentre prospective studies. This review underscores EUS as a transformative tool in unraveling IAP’s enigma and advancing diagnostic and therapeutic strategies.

## 1. Introduction

Acute pancreatitis (AP) is defined as idiopathic (IAP) when the aetiology is unclear after a full clinical assessment and comprehensive diagnostic investigation, which includes imaging exams such as transabdominal ultrasound (US) and computed tomography (CT) and a comprehensive panel of laboratory tests comprising calcium and triglycerides levels [1].

The most common causes of AP are gallstones, alcohol, chronic pancreatitis, and other pancreatic parenchymal, ductal, and ampullary disorders, followed by pancreatic neoplasms along with hypertriglyceridemia, hypercalcemia, and drug-related pancreatitis. The aetiology of AP remains unclear in approximately 10–30% of patients, and these cases are defined as IAP [2,3,4,5,6,7,8,9]. Determining the aetiology of AP can be challenging, especially in those patients who do not have a significant history of alcohol use and in those who do not exhibit evidence of gallstone disease. During the initial workup, several causes for pancreatitis may be missed despite a timely diagnostic approach with conventional imaging techniques and routine laboratory tests. Therefore, in clinical settings of unexplained AP, advanced imaging techniques and endoscopic procedures are often considered.

Although not always used primarily for diagnostic and therapeutic purposes in biliary disorders, EUS remains the cornerstone of the diagnostic and staging algorithm for various lesions of the gastrointestinal tract and has evolved as one of the most accurate imaging options in the evaluation of several pancreatic diseases [6,7,8,9]. In prospective studies, EUS has been shown to reliably identify the cause in up to 79% of patients after a single episode of AP. Commonly detected etiologies include microcholedocholithiasis, biliary sludge, chronic pancreatitis, or small pancreatic tumours undiscovered on cross-sectional imaging, and for this reason, EUS is usually recommended in individuals of 40 years of age or older with AP and no identifiable aetiology [1,10]. There is no definitive data on the risks and benefits of immediate endoscopic examination in the evaluation of AP when no causative aetiology is directly identified. In accordance with current guidelines, patients with possible IAP should be referred to centres of expertise for pancreatic diseases to enable a more accurate etiological investigation [10,11]. 

In the present study, we aimed to review the role of EUS in the diagnosis of IAP, discussing the accuracy, timing, and safety of EUS in different clinical settings and the possible future perspectives derived from recent technological advances and improved endoscopic devices. In the present study, we investigate the role of EUS in identifying the actual cause of IAP by examining the available literature. The goal of this study is to assess possible existing evidence regarding EUS accuracy, timing, and safety and explore potential trends. A discussion of future technological advancements in EUS for diagnostic purposes is also included.

## 2. Materials and Methods

This review was performed in accordance with the Preferred Reporting Items for Systematic Reviews and Meta-Analysis (PRISMA) guidelines. A systematic literature search was conducted to identify relevant studies for this review. An online search was conducted using PubMed, Embase, Web of Science, and the Cochrane Library. Articles were comprehensively searched using appropriate keywords and MeSH terms related to the use of EUS for identifying the actual cause of IAP; search terms included the following: “idiopathic acute pancreatitis” AND “endoscopic ultrasound” OR “endoscopic ultrasonography” OR “EUS” OR “endosonography” OR “echoendoscopy”. The search was limited to articles published in English from the initial reports of the diagnostic application of EUS in the clinical setting from 1995 through April 2023.

All prospective and retrospective studies and systematic reviews that involved patients diagnosed with IAP and utilized EUS as a diagnostic modality were considered for inclusion. Letters, editorials, short reviews, and conference abstracts were excluded. Data extraction was independently performed by four authors (CF, RL, PA, AA) using a standardized data extraction form. Any discrepancies or disagreements in data extraction were resolved through discussion and consensus among the four authors who performed the search.

## 3. Results

From a total of 302 studies, 60 clinical reports were finally included in this review (Figure 1).

### 3.1. Role of EUS in IAP

EUS is a minimally invasive procedure that combines endoscopy and high-frequency ultrasound to provide direct visualization of the pancreas and the surrounding anatomical structures, including the biliary system, the Vater papilla, the pancreatic duct, and the duodenal wall. EUS-guided fine-needle aspiration and biopsy (EUS-FNA and FNB) further enable sampling of pancreatic tissue and fluid, aiding in the diagnosis of pancreatic diseases. EUS can also be performed in conjunction with Endoscopic Retrograde Cholangiopancreatography (ERCP) to treat the conditions diagnosed using EUS. EUS is a specialized procedure that requires skilled endoscopists with experience in pancreatic imaging and interventions. It is essential to have access to a well-equipped endoscopy unit and experienced medical professionals for the procedure [9,10,11].

Several important causes of AP initially diagnosed as idiopathic can be identified through EUS. These include microlithiasis in almost 30–40% of patients; other aetiologies potentially detectable using EUS are small common bile duct stones, *pancreas divisum*, small pancreatic tumours, pancreatic duct strictures, and sphincter of Oddi dysfunction [12,13,14,15,16,17,18,19,20]. EUS has also been useful in identifying subtle structural abnormalities, such as choledochal cysts, cystic dystrophy of the duodenal wall, and ampullary stenosis, which may contribute to pancreatitis development and pancreatic parasites [21,22]. Additionally, EUS-FNA can be used to support the diagnosis of autoimmune pancreatitis, ruling out malignancy and guiding appropriate management [23,24].

EUS was confirmed in all studies included in this review as a valuable tool for evaluating patients with IAP, where the exact cause could not be determined after a comprehensive clinical, laboratory, and imaging workup. While MRI and CT can identify most of the pancreatic parenchyma and duct abnormalities, they usually do not accurately detect microlithiasis. Current IAP/APA (International Association of Pancreatology and American Pancreatic Association) guidelines suggest that EUS should be performed in IAP even after the first episode, as it can identify the aetiology and potential complications of AP [10]. EUS and MRCP should both be used in the diagnostic workup of IAP. EUS has higher diagnostic accuracy in the etiological diagnosis of IAP, whereas MRCP or secretin-enhanced MRCP (S-MRCP) are superior to EUS in diagnosing a possible anatomic alteration in the biliopancreatic duct system [25].

### 3.2. Optimal Timing for EUS in IAP

In all the studies, EUS is never recommended as a first-line investigation but rather as a second-line screening procedure or as a follow-up procedure when the initial workup is inconclusive. In particular, EUS is indicated when conventional imaging studies, such as abdominal ultrasound and CT, fail to identify the cause of AP [10,25]. 

There is no specific optimal timing for performing EUS after the onset of AP. However, it is generally recommended to perform EUS after 4 weeks in cases of mild–moderate acute pancreatitis when the acute phase has resolved; this is suggested to decrease the risk of potential complications of the procedure. Furthermore, the reduced inflammation and oedema of the pancreatic parenchyma after 4 weeks from the initial observations can enable better visualization and assessment of pancreatic lesions. The optimal timing for EUS should, however, be determined based on the severity of radiological pancreatitis, specifically the CT severity index, where a 6-week interval appears to be safe in the case of severe pancreatitis [26]. 

In some selected patients with recurrent IAP, it is possible to anticipate EUS, particularly in those with suspected biliary obstruction or in those with mild–moderate pancreatitis, with a high likelihood of having structural abnormalities not identified using standard imaging [27,28]. 

### 3.3. Role of EUS in Biliary Pancreatitis

Recent data suggested that with EUS, a biliary aetiology could be established in 37% of IAP patients [23,25] (Figure 2). When a biliary cause is found, it should be treated with ERCP, laparoscopic cholecystectomy (LC), or both. Some centres even recommend empiric LC in patients after single or recurrent attacks of IAP due to possible occult biliary disease [29,30]. There is an association between elevated ALT levels and acute biliary pancreatitis, with a positive predictive value of 85% for ALT > 150 U/L within 48 h after the onset of symptoms. Therefore, elevated ALT levels in IAP are strongly suggestive of a biliary aetiology [31,32,33]. Even if LC could be beneficial in such cases where the cause was microlithiasis or biliary sludge that was not identified, EUS can rule out other rare causes of AP. A thorough investigation of the patient’s biliary anatomy with MRI and EUS could identify rare conditions that could cause a pancreatitis recurrence or some conditions that if left untreated would give the patients higher morbidity and mortality, such as chronic pancreatitis, *pancreas divisum*, pancreatic neoplasm, cystic neoplasm, IPMN, pancreatic duct stones, pancreatic duct strictures, and other anatomic abnormalities [25]. IAP has indeed a relatively high recurrence rate, up to 25% during the 3 years after the first episode [34]. Chronic pancreatitis seems to be more frequent in patients with recurrent IAP, and it could be the manifestation of progressive organ damage from recurrent episodes of IAP [35].

### 3.4. Role of EUS in Idiopathic Acute Recurrent Pancreatitis, Pancreas Divisum, and Sphincter of Oddi Dysfunction

Idiopathic acute recurrent pancreatitis (IARP) is the occurrence of two or more episodes of IAP without concurrent clinical or imaging evidence suggestive of chronic pancreatitis or other diseases. If left untreated, the underlying cause of IARP could lead to chronic pancreatitis [36,37,38,39,40,41,42,43,44,45,46,47]. As LC is often performed in patients with IAP, many patients with IARP have a history of cholecystectomy. Although there is a lower rate of diagnosis of biliary disease in patients without a gallbladder, lithiasis is still the second most common EUS finding in IARP after chronic pancreatitis. Sphincter of Oddi dysfunction and *pancreas divisum* have been associated with high recurrence rates in other studies of IARP [48,49,50,51,52].

Pancreas divisum is a congenital anomaly resulting from the failure of fusion of the ventral and dorsal pancreatic ducts; it is sometimes identified as a potential cause of IAP. EUS offers direct visualization of the pancreatic duct system with a high sensitivity, similar to MRCP [36]. In cases where *pancreas divisum* is associated with pancreatitis, the literature suggests that ERCP with minor papilla sphincterotomy and dorsal duct (Santorini) stent placement can effectively serve as preventive measures against future episodes and may provide relief from symptoms [34,53,54,55,56]. As a treatment tailored according to the aetiology is associated with a reduction of recurrence, an EUS-based management strategy is suggested in patients with IARP [54].

Sphincter of Oddi dysfunction (SOD) encompasses clinical syndromes with biliary and pancreatic manifestations. Biliary SOD commonly follows cholecystectomy, while pancreatic SOD relates to IARP [57,58]. The revised Milwaukee Biliary Group classification can assist in diagnosing and categorizing SOD into three types. Type I exhibits biliary-type pain, abnormal liver function test results, and a dilated bile duct. Type II involves biliary-type pain with one laboratory or imaging abnormality, while Type III involves recurrent biliary-type pain alone [59]. Manometric evidence of SOD varies among patients. Treatment of Type I SOD involves ERCP with biliary sphincterotomy. Type II SOD, with less objective evidence, may require ERCP with sphincterotomy guided by sphincter of Oddi manometry (SOM) or empiric biliary sphincterotomy. The relationship between manometric findings, disease aetiology, and response to therapy remains unclear, and empiric sphincterotomy is considered an alternative. Recent trials suggest limited benefits of ERCP and sphincterotomy for Type III SOD. Pancreatic SOD predisposes a patient to recurrent acute pancreatitis, and sphincterotomy can reduce its frequency, though recurrence rates remain significant. Temporary pancreatic stenting during sphincterotomy reduces procedure-induced pancreatitis risk [57,58]. In conclusion, SOD is a complex condition with different subtypes and management strategies. The role of SOD and medical therapy efficacy remain uncertain [59].

### 3.5. Role of EUS in Pancreatic Tumours and Autoimmune Pancreatitis

IAP can be associated with pancreatic tumours, although they account for a small percentage of cases. Whether benign or malignant, pancreatic tumours can contribute to AP through various mechanisms. When a tumour is in the head of the pancreas, it can obstruct the pancreatic duct or the common bile duct, impairing the drainage of pancreatic enzymes and triggering inflammation. Sometimes, tumours can directly induce local inflammation and disrupt normal pancreatic tissues, leading to pancreatitis development. It is important to note that pancreatic tumours associated with IAP are relatively uncommon, only between 2 and 5% of cases. However, given the implications for patient management and prognosis, a pancreatic tumour should be excluded in patients with IAP, especially older individuals, or those with risk factors for malignancy [60,61,62]. The same diagnostic imaging techniques used to evaluate pancreatitis, such as CT, MRCP, and EUS, are used to detect the presence of pancreatic tumours [41,63]. Pancreatitis secondary to obstruction of the pancreatic duct from a tumour is more likely to be mild and recurrent, as ductal obstruction is usually partial [64].

It should not be forgotten that autoimmune pancreatitis, which is a rare form of chronic pancreatitis that can sometimes present acutely, can form tumour-like masses or duct strictures, especially in pancreatic involvement of IgG4-related disease (IgG4-RD). As IgG4-RD generally presents with other clinical features such as retroperitoneal fibrosis, nephritis, thyroiditis, sclerosing cholangitis, sialadenitis, and interstitial pneumonia, testing for IgG4 serum concentration is suggested when IAP is associated with any of these signs [65]. Autoimmune pancreatitis is more common in the elderly, and up to 50% of these patients are diagnosed with a distant malignancy within 1 year of the pancreatitis episode, especially gastric, lung, or prostate carcinoma. This association could suggest that autoimmune pancreatitis may represent a paraneoplastic syndrome [66].

To summarize, while tumours and autoimmune diseases are not frequent causes of IAP, they should be considered in the diagnostic evaluation, particularly in older individuals or those with additional risk factors. Pancreatic tumours can cause pancreatitis because of their location or for some local physiological effects; timely recognition and appropriate management of pancreatic tumours associated with pancreatitis are essential for optimizing patient outcomes. Autoimmune pancreatitis and IgG4-RD can sometimes mimic tumours by creating masses or may possibly be associated with the presence of a distant tumour. Lastly, it is important to remember that certain tumours can result in secondary pancreatitis due to hypercalcemia. These tumours include multiple myeloma, parathyroid tumours, leukaemia, and small-cell lung cancer [11,16].

### 3.6. Safety and of EUS

EUS generally has a low complication rate [67]. Serious complications such as oesophagealor duodenal perforation are extremely rare but higher than those observed with conventional endoscopy due to the rigid linear US transducer mounted on the tip of the echo-endoscope. Additionally, when EUS is used to guide therapeutic interventions or to obtain fine-needle aspiration biopsies to examine suspicious pancreatic lesions, some post-procedural bleeding episodes have been described. All reports dealing with safety are in accordance with the fact that the experience of the endoscopist performing the procedure can impact its safety. Performance of EUS by experienced specialists can minimize potential complications and increase the likelihood of obtaining accurate diagnostic information. EUS is considered a valuable and safe diagnostic tool even in children [49,68].

### 3.7. Diagnostic Accuracy

The primary challenge in evaluating EUS accuracy in detecting the actual cause of IAP is the absence of an independently established gold standard test for diagnosing IAP possible etiologies. Unlike some medical conditions where a definitive test or criterion is available to establish accuracy, IAP is inherently complex and heterogeneous. Various factors can contribute to IAP, including microlithiasis, small bile duct stones, pancreas divisum, congenital pancreatic duct abnormalities, small pancreatic tumors, and many others. These etiologies often require different diagnostic approaches. The diagnostic yield of EUS in IAP may vary depending on the specific cause and the specific expertise of the medical professionals involved. In the absence of a gold standard test for IAP, it becomes exceedingly difficult to definitively confirm the accuracy of EUS in detecting its causes. Despite these limitations, EUS accuracy has been reported in a review article considering 34 studies in comparison with MRCP where EUS seems advantageous compared to MRCP with a diagnostic yield of 64% vs. 34% [36]. However, the diagnostic landscape of EUS in the general setting of IAP relies on clinical judgment, exclusion criteria, radiological imaging, and endoscopic techniques like EUS, and each of these aspects has some advantages and limitations.

### 3.8. Recent Advances in Technology and Improved Endoscopic Devices

The incorporation of advanced imaging modalities, such as contrast-enhanced EUS, intraductal US, and elastography, may provide additional information about tissue characteristics and vascularity, aiding in the differentiation of benign and malignant lesions [69,70]. This could help to identify the specific cause of IAP in some cases.

EUS-guided fine-needle aspiration (EUS-FNA) and fine-needle biopsy (EUS-FNB) have proven valuable in obtaining tissue samples for pathological analysis. Future advancements in interventional EUS techniques could potentially allow for real-time on-site evaluation of obtained tissue samples, enabling a more rapid and accurate diagnosis of the underlying cause of pancreatitis.

With ongoing advancements in molecular and genetic testing, EUS-guided acquisition of tissue samples could facilitate targeted analysis of specific genes and molecular markers associated with pancreatitis. This personalized approach may lead to a better understanding of the underlying mechanisms of IAP and potentially aid in tailoring treatment strategies.

The integration of AI and machine learning algorithms into EUS imaging analysis in the future could help endoscopists detect subtle abnormalities and patterns that might be missed by the human eye [71]. AI-driven diagnostic support may lead to earlier and more accurate identification of the cause of AP, especially in cases where the aetiology is challenging to determine.

## 4. Discussion

AP is a common gastrointestinal disease characterized by acute inflammation of the pancreatic gland. Its incidence varies between 4.9 and 73.4 cases per 100,000 [1].The diagnosis of AP requires the presence of at least two criteria: typical abdominal pain, high serum lipase or amylase levels, and radiological imaging (US, CT, or MRI) consistent with AP [10,72]. The majority of patients with AP show a mild-to-moderate disease course, but up to 20% of them will develop acute severe necrotizing pancreatitis, which has a mortality rate ranging from 10 to 20% of cases [73,74,75]. In the present review, we aimed to report available data concerning the indication, role, accuracy, timing, and safety of an early EUS in patients initially diagnosed with IAP.

EUS has shown significant promise in the diagnosis and management of various gastrointestinal disorders, including IAP, and some recently improved endoscopic devices can also increase the diagnostic yield of EUS in IAP. Enhanced Resolution and Imaging Capabilities with better image resolution and visualization of the pancreas and surrounding structures allow for more accurate identification of abnormalities, such as pancreatic duct strictures, stones, and tumours, which could be potential causes of AP.

The identification of the underlying cause of IAP has significant clinical implications. It enables targeted therapeutic interventions tailored to the specific aetiology, potentially preventing recurrent episodes and disease progression. EUS aids the decision of the appropriate treatment, such as sphincterotomy, stone extraction, or stent placement, and allows addressing biliary pathologies [50,76,77,78,79]. Additionally, EUS facilitates early detection of neoplastic lesions, leading to timely diagnosis and appropriate treatment, ultimately improving patient outcomes. While EUS has proven valuable in uncovering the cause of IAP, it has some limitations. The procedure requires specialized expertise and may not be readily available in all healthcare settings. Furthermore, rare or less common causes may still remain undetected even with EUS evaluation. Future research should focus on optimizing EUS techniques, exploring the role of advanced imaging modalities, and conducting prospective studies to establish the cost-effectiveness and long-term benefits of EUS-guided management strategies.

Apart from its diagnostic role, EUS has also shown potential as a therapeutic tool [80]. For instance, EUS-guided drainage of pseudocysts or biliary duct strictures could offer a less invasive approach to managing certain causes of pancreatitis [76,81]. As therapeutic EUS techniques evolve, they may complement the diagnostic process and improve patient outcomes.

In conclusion, EUS has revolutionized the evaluation of IAP by revealing previously undetectable underlying causes. Through its ability to provide detailed imaging of the pancreas and adjacent structures, coupled with EUS-guided sampling techniques, it has improved diagnostic accuracy and therapeutic decision-making. Incorporating EUS into the diagnostic workup of IAP allows for targeted interventions, optimizing patient management, and improving outcomes.

The limitations of this review include the potential for publication bias, as only studies published in English were included. The reliance on existing literature and the subjective nature of narrative synthesis may introduce inherent biases. Additionally, the inclusion of studies with varying quality levels and the possibility of selective reporting of outcomes may impact the overall findings. More research is needed to establish the direct link between some new trends in EUS technological advancements and their impact on IAP diagnosis and clinical outcomes.

## Figures and Tables

**Figure 1 diagnostics-13-03256-f001:**
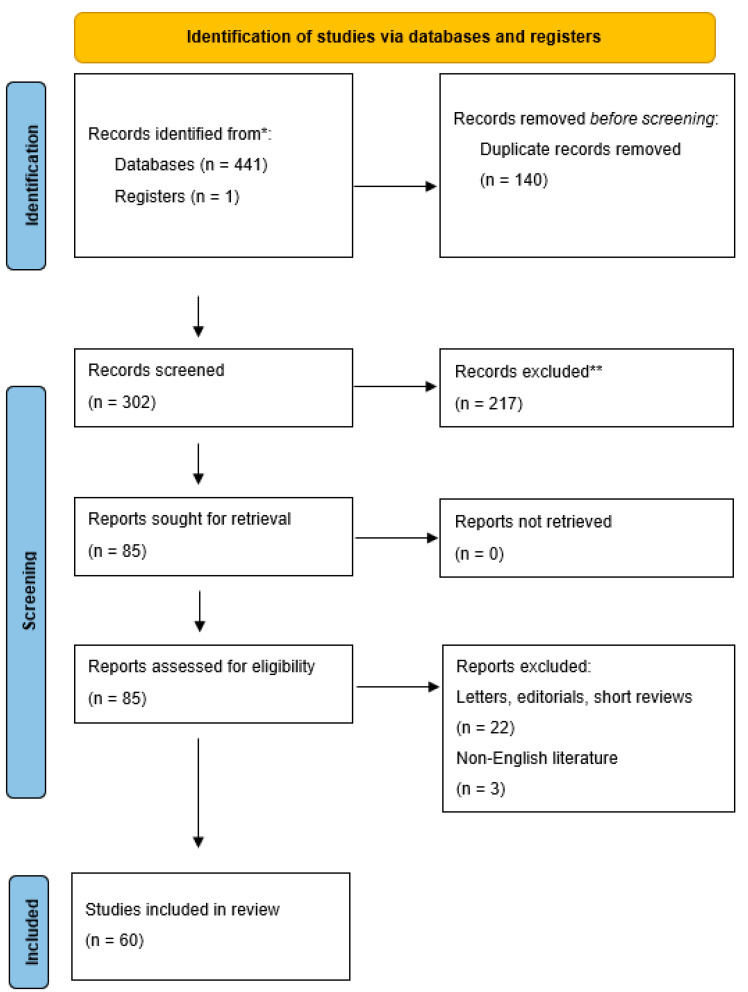
PRISMA flow diagram illustrating the number of records identified (*), included, and excluded (**), and the reasons for exclusions.

**Figure 2 diagnostics-13-03256-f002:**
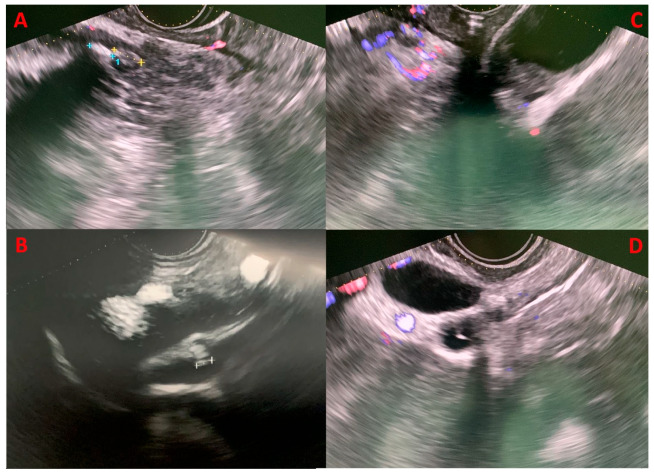
Four clinical cases of biliary microlithiasis without retrodilatation of the common biliary tract, detected using EUS that were not identified in previous diagnostic examinations (transabdominal US, CT, and MRCP). Panels (**A**,**B**) display microcholedocholithiasis in the intrapancreatic and prepancreatic segments, respectively. Panel (**C**) highlights the presence of gallbladder sludge, while panel (**D**) demonstrates microlithiasis in the cystic duct stump.

## Data Availability

The data that support the findings of this study are available in PubMed, Embase, Web of Science, and the Cochrane Library (see Section 2).

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
