# Peer review of "Endoscopic Ultrasound to Identify the Actual Cause of Idiopathic Acute Pancreatitis: A Systematic Review"

_diagnostics, 2023, doi:10.3390/diagnostics13203256_

Round 1
Reviewer 1 Report
In this article, the authors review the role of EUS in the diagnosis of IAP.
The authors stated that the aim of this study is to review the role of EUS in the diagnosis of IAP, discussing the accuracy, timing and safety of EUS. However, this article fails to achieve this aim. I do not think this article is up to the level of publication in Diagnostics.
1.The optical timing and safety of EUS in AIP is not discussed scientifically, just describes the authors opinion.
2. The role of EUS in biliary pancreatitis, idiopathic acute recurrent pancreatitis, pancreas divisum, and sphincter of Oddi dysfunction, pancreatic tumors is not well discussed. It does not state what evidence and how useful it is.
3. Recent advances in EUS are described, but the content has little relevance to the diagnosis of IAP.
Author Response
We would like to express our sincere gratitude to the Editor-in-Chief and the Reviewers for considering our manuscript entitled "Endoscopic Ultrasound to Identify the Actual Cause of Idiopathic Acute Pancreatitis: A Comprehensive Review". We appreciate your feedback and the opportunity to address your concerns and suggestions regarding our article. We have carefully considered your comments and we have revised the manuscript according to reviewers' suggestions. Red marks in the revised manuscript indicate the changes. The following step-by-step responses are provided in response to the reviewers' comments.
Reviewer n. 1
In this article, the authors review the role of EUS in the diagnosis of IAP. The authors stated that the aim of this study is to review the role of EUS in the diagnosis of IAP, discussing the accuracy, timing and safety of EUS. However, this article fails to achieve this aim. I do not think this article is up to the level of publication in Diagnostics.
Q. n.1."The optical timing and safety of EUS in AIP is not discussed scientifically, just describes the authors opinion."
Response to Q n. 1. Thank you for your feedback and valuable comments regarding our article. We greatly appreciate your constructive criticism, which has allowed us to clarify some points and enhance our article's scientific quality.
- Optimal Timing for EUS in IAP: We acknowledge your concern regarding the optimal timing for EUS in IAP. You rightly point out that we relied on expert opinions. In the absence of robust prospective studies to recommend optimal timing we have identified only one retrospective study that provides some general but valuable insights into this matter (Ref. 26). The rationale behind performing EUS is suggested to be at least 4 weeks after the initial presentation of mild –moderate acute pancreatitis since inflammation and oedema within the pancreatic parenchyma can prevent accurate visualization and assessment of pancreatic lesions during the acute phase. The same study suggests that the optimal timing for EUS should be based on radiological pancreatitis severity, specifically according to the CT severity index and conclude that an interval greater than 6 weeks appears to be safe before performing EUS among patients with a high CT severity index. We thank you for pointing out this valuable issue and we included this specific observation in our revised manuscript.
- Safety of EUS in IAP. Regarding your concern about EUS safety in IAP: our initial review indeed revealed limited specific data on EUS safety specifically in the setting of IAP when the acute phase of pancreatitis can increase the risk in comparison to EUS for other indications. We have included this observation in our revised manuscript to make readers aware of the current gaps in the literature. Furthermore, we conducted a comprehensive search to identify if any new studies or relevant data that have emerged since our initial review. It was possible to find relevant data regarding EUS safety only in a more general contest which we included as an additional reference. No data were found about safety in the psecific clinical setting of IAP.
-Accuracy of EUS on IAP.
We are aware of your concerns regarding the accuracy of EUS in identifying the actual cause of idiopathic acute pancreatitis (IAP), and we understand that you are seeking accuracy data. However, we would like to clarify that assessing the accuracy of diagnostic tests, such as EUS in the context of IAP, presents a unique challenge. In the setting of IAP, EUS accuracy is limited by the lack of a gold standard test for diagnosing possible causes. Unlike some medical conditions where a definitive test or criterion is available to establish accuracy (gold standard test), IAP is inherently complex and heterogeneous. These aetiologies often require different diagnostic approaches and may not always present with clear-cut clinical or radiological findings. For each different possible aetiologies and the diagnostic yield of EUS in IAP may vary depending on the specific cause and the specific expertise of the medical professionals involved. It is extremely difficult to definitively confirm the accuracy of EUS in detecting possible aetiologies without a recognized gold standard test. We found that EUS accuracy has been reported in a review article (ref. n. 36) in comparison with Magnetic Resonance Cholangio-pancreatography (MRCP) where EUS seems advantageous compared to MRCP in the detection of a possible biliary disease. However, the diagnostic landscape of EUS in the general setting of IAP relies on clinical judgment, exclusion criteria, radiological imaging, and endoscopic techniques like EUS, and each of these aspects has some advantages and limitations. Our review aimed to consolidate the available evidence and provide a comprehensive assessment of whether EUS is a valuable diagnostic tool. To address this inherent limitation and the reviewer's concern, we have highlighted in our article the unavailability of reliable accuracy data in IAP. We have highlighted the importance of further research and we have emphasized in the revised manuscript this concept and the need for multicenter prospective studies to explore the refinement of EUS techniques in relation to its accuracy. We would like to stress that incomplete reporting of comparative accuracy studies is a general problem in the medical literature. There is a clear need for more informative research reports, which could be facilitated through the development of explicit guidelines specifically developed as a guide for comparative accuracy studies. Please accept our sincere appreciation for your feedback and understanding that you are concerned about the lack of accuracy evaluations. Due to the current state of knowledge and complexity of IAP, our review can only provide valuable insights into EUS's role in diagnosing IAP causes.
Q. n. 2 The role of EUS in biliary pancreatitis, idiopathic acute recurrent pancreatitis, pancreas divisum, and sphincter of Oddi dysfunction, pancreatic tumors is not well discussed. It does not state what evidence and how useful it is.
Response to Q. n 2 Thank you for your feedback. We appreciate your input, and we would like to address your concerns. It would be worthwhile to clarify the scope and purpose of our review article. We would like to point out that the scope of the article did not aim to discuss extensively the role of EUS in multiple varying causes of pancreatitis such as biliary pancreatitis, idiopathic acute recurrent pancreatitis, pancreas divisum, sphincter of Oddi dysfunction, and pancreatic tumors. Our primary aim was to review the role of EUS in the detection of the actual cause of IAP. Our focus was deliberately narrow to provide a detailed analysis of the EUS contribution to identifying multiple underlying causes of IAP. While we understand the importance of discussing EUS in the setting of different pancreatitis aetiologies, including those you mentioned, such as biliary pancreatitis, pancreas divisum sphincter of Oddi dysfunction and pancreatic tumors, our decision to limit the scope of this review was intentional. These topics warrant separate, dedicated reviews, and our goal was to provide a focused and in-depth analysis of EUS in IAP. We appreciate your feedback, and we hope our clarification regarding the article's scope aligns with your understanding.
Q. n. 3 Recent advances in EUS are described, but the content has little relevance to the diagnosis of IAP.
Response to Q. n. 3
Recent Advances in EUS: Our article discusses recent advances in EUS, and you are right to note that their direct relevance to IAP diagnosis may not be fully proven. We recognize that these advances are still evolving and may not have specific data showing their impact on IAP diagnosis yet. However, we would like to stress that some evolving aspects of these technologies, such as artificial intelligence (AI) may have clinical relevance. We have clearly stated in the paragraph that “The integration of AI and machine learning algorithms into EUS imaging analysis in the future could help endoscopists detect subtle abnormalities and patterns that might be missed by the human eye [71]. AI-driven diagnostic support may lead to earlier and more accurate identification of the cause of AP, especially in cases where the aetiology is challenging to determine”. This paragraph aims to highlight the broader context in which EUS is evolving as a more accurate diagnostic tool for pancreas diseases”. We thank you for pointing out this valuable issue and we included for a better clarification the following sentence to Discussion "More research is needed to establish the direct link between some new trends in EUS technological advancements and their impact on IAP diagnosis and clinical outcomes".
Reviewer 2 Report
Just two small corrections in the text
Line 31 "it boasts minimal safety concerns, and is crucial to prevent complications", CHANGE ïƒ EUS, when conducted by experienced endoscopists boasts minimal safety concerns and can be crucial to prevent complications.
Line 252 intraductal EUS, CHANGEïƒ intraductal US
Only minor editing required
Author Response
Reviewer n 2.
Dear Editor-in-Chief and dear Reviewers, on behalf of all the authors, I would like to express our sincere gratitude for considering our manuscript entitled "Endoscopic Ultrasound to Identify the Actual Cause of Idiopathic Acute Pancreatitis: A Comprehensive Review". We appreciate your feedback and the opportunity to address your concerns and suggestions regarding our article. We have carefully considered your comments and we have revised the manuscript according to the reviewers' suggestions. Red marks indicate the changes in the revised manuscript. The following step-by-step responses are provided in response to the reviewers' comments.
Reviewer n. 2
Just two small corrections in the text.
Line 31 "it boasts minimal safety concerns, and is crucial to prevent complications", CHANGE à EUS, when conducted by experienced endoscopists boasts minimal safety concerns and can be crucial to prevent complications.
Response to Q. n.1
Thank you for your suggestion. We have changed the sentence in “With a low rate of complications, EUS poses minimal safety concerns”.
Line 252 intraductal EUS, CHANGEà intraductal US
Response to Q. n.1
Thank you. We have edit with “intraductal US”
